# LiveSchema: A Gateway to Relational Data Analysis and Transformation

Mattia Fumagalli[1], Marco Boffo[2], and Fausto Giunchiglia[2]

[1] Conceptual and Cognitive Modeling Research Group (CORE),
Free University of Bozen-Bolzano, Bolzano, Italy
{mattia.fumagalli@unibz.it}
[2] Department of Information Engineering and Computer Science (DISI),
University of Trento, Italy
{marco.boffo@studenti.unitn.it,fausto.giunchiglia@unitn.it}

**Abstract.** One of the major barriers to the training of statistical models on *knowledge representations* is the difficulty that scientists have in finding the best input data to be used for addressing their prediction goal. In addition to this, a key challenge is to determine how to manipulate these *relational data*, which are often in the form of particular triples (i.e., *subject*, *predicate*, *object*), to enable the learning process. This paper describes the *LiveSchema* initiative, namely a gateway that leverages the gold-mine of relational data collected by many existing *ontology catalogs*. By implementing a continuously updating aggregation facility, LiveSchema aims at providing a family of services that can be used to easily access, accurately analyze, transform and re-use data in a machine learning scenario.

**Keywords:** Relational data · Ontologies · Ontologies catalog · Ontologies analysis and transformation · Relational learning

## 1 Introduction

Finding the best data to train statistical models and properly address the target learning goal is widely recognized as one of the most pivotal tasks in *Machine Learning (ML)* [4]. ML models highly depend, indeed, on the quality of data they receive as input. While, so far, the development of highly efficient and scalable learning methods, to address key prediction tasks (see, for instance, *image classification* and *information patterns recognition* [1]), helped data scientists and analytics professionals in scaling their activities, the process of finding, selecting and improving the quality of these data still requires a huge amount of time and manual effort [9]. This latter challenge is also present when statistical models are trained on *knowledge representations*, like *knowledge graphs* and *ontologies* [10], where the data received as input are graph-structured data, consisting of *entities* (or nodes) and labelled links, or *edges*, (relations between entities). In this setting, the final learning goal can be identified as the *prediction of missing*

*relations between nodes*, the *prediction of nodes properties*, and the *clustering of the nodes based on their connections*, these being common goals arising in many scenarios, such as analysis of social networks and biological pathways [11]. Consequently, the efficacy of the ML algorithms directly depends on the quality of the input graphs, as well as their relevance to the domain of application.

In this paper, leveraging the ideas presented in [6,7,8] and [5], where an approach to analyse relational knowledge representations to address *Entity Type Recognition (ETR)* tasks has been devised, we introduce the *LiveSchema* initiative, namely a gateway that aims at providing a general support for data scientists in the analysis, transformation, and (re-)use of relational data like ontologies. The platform is accessible at `http://liveschema.eu/` and it is ready to be demonstrated. The admin functionalities can be accessed and tested at `http://liveschema.eu/user/login`, by using 'reviewer' as *admin/password*.

The paper is organized as follows: Section 2 introduces the motivation grounding the LiveSchema initiative. Section 3 gives a brief overview of the current state of the project.

## 2   Motivation

As an example scenario, suppose that a data scientist needs to run a standard *Entity Type Recognition task (ETR)*[3], as it is described in [8] and [12], where the goal is to recognize objects of the type 'Person' across a set of multiple tabular data, coming, for instance, from an open data repository. This may involve that she needs to find a reference ontology containing: *i.* the target class and corresponding label; *ii.* possibly a huge number of properties for the target class, in order to increase the probability to match some of the input test properties; *iii.* possibly a low number of overlapping properties, in order to decrease the number of false-negative/positive predictions.

The process of searching, analyzing, and transforming the target ontology can take a long time and it may involve a considerable effort. The scientist has, indeed, to go through a broad search over the available resources and related catalogs, possibly checking multiple data versions and formats. Moreover, once the candidate resources are identified, she should run an analysis of the data, to better understand their reliability in relation to the target task. Additionally, this analysis (see, for instance, the simple data about the number of properties associated with each class) requires a processing phase that is assumed to be set-up and run directly by the scientist. As a final step, if the scientist succeeds in finding the data she needs, a transformation process must be run to re-use the relational data in the reference ETR set-up. *What if the scientist can run all these operations in one single place with the support of ready-to-be-used built-in facilities?*

LiveSchema was precisely devised with this key goal. Firstly, the gateway aims at supporting scientists in better finding the relational data they need. Indeed, by leveraging the updates of some of the best state-of-the-art catalogs,

---

[3] Relational data was proven to be key also in a *transfer learning* setting [2,3].

LiveSchema offers an aggregation service that allows searching and keeping track of the evolution of the knowledge representation development community in one place.

Moreover, by implementing some key state-of-the-art libraries, LiveSchema aims at facilitating the data analysis and preparation process. Most of the implemented libraries, indeed, require an *ad-hoc* set-up and may involve the combination of multiple components and environments, involving some coding and development skills that not all the pure data scientists have. In this sense, LiveSchema aims at offering a platform that unites data analysis, data processing, and machine learning model deployment, making them easily accessible, reusable, and less time-consuming.

## 3  Current State

At the current state LiveSchema relies on three main state-of-the-art catalogs, namely LOV, DERI, FINTO[4] and a *custom catalog*[5], where some selected resources are stored.

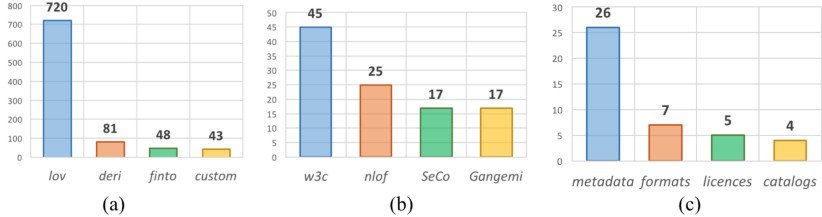

Fig. 1: LiveSchema overall statistics.

From each catalog, multiple data sets have been scraped and uploaded in an automated way. The distribution of data sets for each catalog is represented by the chart .*a* in Figure 1. Each catalog is associated with a *provider* (1.*b*). From all the catalogs, as from 1.*c*, 26 types of metadata have been scraped (e.g., *'name'* and the *'URL'*). Along with the metadata scraping, a cross-check is performed to ensure that LiveSchema is not breaking any license agreement. Currently, 5 kinds of licenses are admitted. All the stored data sets are then serialized into multiple formats (including `RDF` as native format and `FCA` format to enable embedding and analysis).

The overall architecture of LiveSchema is depicted in Figure 2 below. Five are the main components. The (1) *User Interfaces (UIs)* and (2) the *APIs*, provided by the CKAN[6] platform, and that we partially customized according to our set-up needs, provide the main accesses to the LiveSchema environment. The (3) *stoking components* and the (4) *forging components* cover the main novel contributions of the LiveSchema initiative and offer the possibility to harvest, generate, and process data. The (5) *storage* allows collecting in one place all the data collected by other catalogs, provided by the users or generated through the

---

[4] `https://lov.linkeddata.es/`, `http://finto.fi/en/`, `http://vocab.deri.ie/`

[5] `https://github.com/knowdive/resources/blob/master/otherVocabs.xlsx`

[6] `https://docs.ckan.org/en/2.8/contributing/architecture.html`

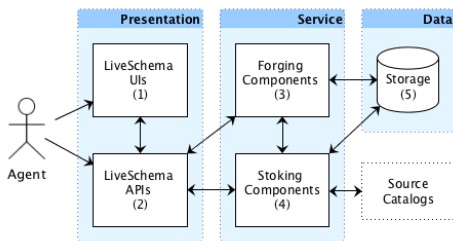

Fig. 2: Overview of the LiveSchema architecture.

services. All these components are grouped in three main layers: the *i. presentation layer*, which enables a community of users (administrators, editors, and standard users) in maintaining the whole portal and its applications; the *ii. service layer*, where the process of data aggregation and collection (stoking), and the data processing services (forging) are enabled; and the *iii. data layer* where data are stored.

The possibility of extending LiveSchema with new source catalogs, data sets, and services is already implemented. Moreover, the identification of new catalogs to be used as additional sources is part of the immediate future work.

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
