# OpenReview forum: "LiveSchema: A Gateway to Relational Data Analysis and Transformation"
_eswc-conferences.org/ESWC/2021/Conference/Poster_and_Demo_Track — Submitted to ESWC2021 P&D_

### Official Review · AnonReviewer3 · 2021-04-08
**Interesting project but the paper deserves strong improvements**

**Rating:** 6
**Confidence:** 3

**Review:**

This short paper describes a project meant to support data scientists in the task of searching and transforming some selected vocabularies to serve as input of downstream ML applications.
The paper refers to a web portal based on the CKAN framework. The portal is functional, hosts a copy of multiple data portals, and provides an entry point for various running processing.

Although the demonstration portal seems convincing in the sense that the work has been done and is usable, the paper remains very unclear with respect to what is actually provided by the authors, what services are available and how/where there can be executed.
The paper presents the project as a "gateway". Since it hosts data, I understand this is a gateway to ML algorithm executable from other platforms, is that the case?

Firstly, the paper is not said whether this submission is for a poster or a demo (this is not mentioned on the OpenReview portal either). I would assume a demo given the existing portal.

Section 2 starts with an example scenario. If this is what you intend to demonstrate, then please make this much clearer and be more specific.
Furthermore, the paper frequently refers to "relational data". With my background, I understand "tabular data" or "data from a relational database". Yet, the authors use the expression for any kind of ontologies.
The term "data" is used several times although I'm not sure whether we speak of the tabular data to be processed (the ETR task), or the candidate reference ontology that has been selected.

Section 3 almost exclusively focuses on the data part. It reports statistics about the data that was imported in the portal, and briefly sketches the architecture. But no information whatsoever about the services: what happens if I run the knowledge embedder? Is this deployed on a cluster provided with the portal? Is this executed on a third-party platform somewhere? Where?

Finally, the paper needs to mention at least a few related works. For instance, how would you compare to OpenML or similar projects?


-- Minor remarks.

The DERI vocabularies website has been unavailable for quite some time now, how/where did you get them?

"LiveSchema offers an aggregation service that allows searching and keeping track of the evolution of the knowledge representation development community in one place.": this sentence is quite confusing, does the p/f keep track of changes on the source data portals? Or are you referring to KR methods and practices?


**Anonymity:**

Yes, I would like my review to remain anonymous.

---

### Official Review · AnonReviewer4 · 2021-04-14
**Need for more information ont the contribution**

**Rating:** 4
**Confidence:** 3

**Review:**

The paper describes the aggregation site on schema definitions in order to accelerate data usage by scientists. It collects schema definitions from mainly five sites and makes them searchable. The service itself is convenient and useful. But the reviewer wondered whether any innovative or creative efforts are embedded for building the service. Although the authors mentioned in the explanation of the system as
"(3) stoking components and the (4) forging components cover the main novel contributions of the LiveSchema initiative and offer the possibility to harvest, generate, and process data, " there are no more explanations on them. The paper is not informative for the participants of the conference.

**Anonymity:**

Yes, I would like my review to remain anonymous.

---

### Official Review · AnonReviewer1 · 2021-04-14
**LiveSchema: A Gateway to Relational Data Analysis and Transformation review**

**Rating:** 4
**Confidence:** 3

**Review:**

The paper describes the LiveSchema system, which is a gateway that aggregates data from several repositories like Linked Open Vocabularies, Finnish thesaurus and ontology service and DERI vocabularies.

The systems is based on CKAN and it offers two features that the authors declare are the main novel contributions: Stoking (aggregation and collection) and Forging (data processing). The paper doesn't explain in further details of these components.

There is no link to the source code so it seems it is not available. There is also not a lot of details about the architecture or the technical details of the system.

The authors describe some of the features of the system but it is not clear what does the service offer compared with other services like LOV.

The paper doesn't include any comparison with other systems which I think would be useful. It wouldn't need to have a whole section but at least some paragraph comparing it with, for example, LOV, and describing the relationship with CKAN would be helpful.

Minor details:
- The title and the abstract talk about "relational data" which I think is data with relations like nodes and edges, but the traditional meaning of "relational data" for me is "relational databases". I would recommend the authors to use a different name to avoid confussion.
- The arrows in the architecture diagram in fig. 2 are not very helpful because the are double arrows and have no explanation about what they mean.
- The footnote 4 in page 3 that points to LOV, DERI and FINTO has the URIs of those portals in a different order (LOV, FINTO and DERI).
- The paper contains the user/password of admin which I suppose should be changed in case the paper was accepted. Maybe a different way to provide those credentials would be better?





**Anonymity:**

Yes, I would like my review to remain anonymous.

---

### Decision · Program_Chairs · 2021-04-19

Reject